Cambridge Prisms: Global
Mental Health

## Research Article

Ukraine; refugee; trauma; adaptation; emigration

**Corresponding author:**
Ivan Rektor;
Email: irektor@med.muni.cz

# War and women: An analysis of Ukrainian refugee women staying in the Czech Republic

Marek Preiss[1,2,3] 📷, Monika Fňašková[3], Sofia Berezka[3], Tetiana Yevmenova[3], Radek Heissler[1], Edel Sanders[2], Petra Winnette[5] and Ivan Rektor[3,4]

[1]National Institute of Mental Health, Klecany, Czech Republic; [2]University of New York in Prague, Prague, Czech Republic; [3]Central European Institute of Technology (CEITEC), Centre for Neuroscience, Masaryk University, Brno, Czech Republic; [4]Department of Neurology, St. Anne's University Hospital and Faculty of Medicine, Masaryk University, Brno, Czech Republic and [5]Natama Institute, 1st Faculty of Medicine, Charles University, Prague, Czech Republic

## Abstract

In addition to the loss of life, Russian aggression against Ukraine, which began in February 2022, also brings interpersonal losses resulting from the need to emigrate. Parallel to the fighting men, women bear most of the burden of caring for the family. Using in-depth interviews supplemented by questions about adverse childhood experiences and administration of The Centrality of Events Scale and the PTSD Checklist – PCL-5 with 43 Ukrainian women (18–60 years old), we analyzed adaptation to the situation of emigration and the association of their war and earlier experiences with the level of traumatization. Women were interviewed shortly after emigration to the Czech Republic (3–42 week afterward). High levels of adverse childhood experiences and post-traumatic stress symptoms were found. The war was perceived as a currently negative central event associated with traumatic stress symptoms, and 79% of the sample expressed the opinion that the war had changed them. The results of this study suggest an intertwining of previous life experiences with the current need and ability to adapt.

## Impact statement

War, though a devastating event, presents an opportunity to examine the complexities of human behavior and adaptation. Adaptation takes place in the context of the region, for example, in the way the war is fought, the host country's capabilities and willingness to accept refugees. The Czech Republic has accepted large numbers of Ukrainian refugees after the country had long rejected refugees. Nevertheless, the situation of refugees, and especially of women, is difficult. In addition to the current need to adapt to conditions in the Czech Republic, women are haunted by their own past and ideas about the future. Using a small sample of Ukrainian women, this study uses in-depth interviews to map their perceptions of their past, present and future situations. The adaptation of these and other Ukrainian women and their perception by Czech society is crucial for the future acceptance or rejection of refugees in the Czech Republic. Furthermore, our study contributes to the understanding of the interplay among multiple situational and psychological elements that may increase the risk of post-traumatic stress disorder and maladaptive behavior in the region and beyond.

## Introduction

The present conflict between Russia and Ukraine started with the annexation of Crimea in March 2014 and a new war began in February 2022. The fights occurred mostly in the Eastern and Southern parts of the country, but bombing was widespread in most of the Ukrainian regions. Data show there has been an increase in civilian fatalities and injuries, and a long-term impact on the health of generations of Ukrainians is expected (Haque et al., 2022). Many Ukrainian citizens have left the country after the Russian aggression. The EU activated a temporary protection directive, which is expected to be the scheme used in exceptional circumstances of a mass influx to provide immediate and collective protection to displaced persons and reduce pressure on the national asylum systems of EU countries (European Council, 2023). According to the WHO, as of June 29, 2023, over 20 million people have left the country (World Health Organization, 2023).

The Czech Republic hosts the largest number of Ukrainian refugees per capita of any European country (People in Need, 2023). As of October 31, 2022, 453,725 refugees were registered in the Czech Republic (Regional Refugee Response for the Ukraine Situation, 2023). Similar to other host countries, 80% of Ukrainian refugees in the Czech Republic are women and children because most Ukrainian men aged 18–60 years are banned from leaving the country (International Organization for Migration, 2023). While Ukrainian men mostly stayed in

Ukraine and defended their county, women were left to care for their children and extended family and, in many cases, were forced to emigrate.

Women have an increased vulnerability to the mental health consequences of war (Murthy and Lakshminarayana, 2006). Exiles experience increased stress. For example, just the evacuation itself from battered or besieged cities to other parts of Ukraine often results in mental health problems (Ben-Ezra et al., 2023). The refugees were assisted both by the Czech government and the general population; however, the stress remained and, in many cases, intensified due to the continuation of the war. In addition to worries about their loved ones in Ukraine and material difficulties, they had to adapt to the conditions of their stay abroad, with the uncertain outcome of whether they would be able to return to Ukraine in combination with the need to adapt to the conditions of their host country. Traumatized families with children also pose a challenge for Czech pediatricians (Uhlíř and Škarková Stojanová, 2022).

Research conducted between April and August 2022 showed that most refugee Ukrainian children's English is better than their Czech (Prokop et al., 2022). Up to two-thirds of parents say that their child is not well integrated into Czech children's social groups. More than half of the refugees live in cramped conditions – with less than 6 m$^2$ of living space per person (bedrooms and living rooms). Around 14% of them were planning to return home. The income poverty rate among Ukrainian refugees in the Czech Republic is higher than among the Czech population. Even when humanitarian benefits and the value of housing support are included, over 35% of refugees live within income poverty levels (9–10% among Czechs) (Prokop et al., 2022). At the end of 2022, 90% of Ukrainian refugee children were attending Czech primary schools, significantly more than in June 2022 (55%) (Prokop et al., 2022). On the other hand, the intensity of Czech language instruction has declined since June 2022, with only around 16% of all children aged six and over learning Czech for at least 2 hours per day at school, and the rest learning less or not attending school at all. At the end of 2022, 39% of children were well integrated into a group of Czech children, according to their parents.

Even before the war in Ukraine in 2022, there was a growing trend toward xenophobic populism in the Czech Republic. Xenophobic populists have shown to be effective in polarizing society and can often set the tone for the overall political debate on certain topics (ECRI, 2023). In the past, the groups most affected by racist hate speech in the Czech Republic were Muslims and Roma (ECRI, 2023); however, data regarding Ukrainian refugees are missing. Therefore, we are unable to make a comparative analysis at this time. It could be expected that although very high, the initial unprecedented solidarity may wane and will need to be continuously enhanced.

It is likely that the current level of stress among refugees is related not only to war trauma, but also to current adaptation challenges as well as previous prewar experiences. Additionally, it is possible to predict that the stress caused by war will transmit to future generations. In another research context, it was found that the traumatic symptoms of parents who experienced the Holocaust can result in similar symptoms among second-generation populations, even in those individuals who were not directly exposed to the trauma or did not hear about it (Cohn and Morrison, 2017).

It is known that the most important personality trait behind the adverse effects of traumatization is a heritable predisposition (neuroticism) – a tendency to react with strong emotion to adverse events (Paris, 2000). Individuals with diagnosed PTSD were also likely to have a past history of exposure to adversity prior to the historical event in question. In the case of Ukrainian refugees, it may be related to factors such as the original standard of living, protection levels via the accessibility of the law and the mental health quality of the original families. The extent to which the effects of such a predisposition are activated depends on the cumulative effects of stressors, both recent and current events (e.g., acculturation in the host country, friendliness of the population language facilities) and earlier adversities (e.g., early childhood adverse experiences).

In the context of the impact of the Russian aggression against Ukraine in 2022, the aim of this paper is to (1) describe the experiences of adult female Ukrainian refugees in the Czech Republic, (2) focus on the association of the current war and their earlier experiences with the level of traumatization and (3) determine whether the experiences of the current war can influence adaptation to the host country.

## Methods and analysis

### Participants

The study was conducted at the Central European Institute of Technology (CEITEC), Centre for Neuroscience at Masaryk University in Brno from June to October 2022. Participants were recruited in cooperation with Masaryk University, which approached academics from Ukraine who had found temporary employment at Masaryk University. The recruitment of respondents for the study was done through personal contacts of two Ukrainian psychologists and through social media. The research was approved by the ethics committee at Masaryk University and informed written consent was obtained from all participants.

A total of 43 women were recruited. The demographics of the sample are shown in Table 1. Most of them left Ukraine during March 2022, about a month after the start of the war, which began on February 24, 2022 with Russia's invasion of Ukraine. The majority of respondents (98%) had spent their childhood in Ukraine (only one respondent in Russia). The women who participated in the study came from different regions of Ukraine (Donetsk, Dnipropetrovsk, Zaporozhye, Kyiv, Poltava, Sumy, Kharkiv and Kherson oblasts). All of them had Ukrainian

**Table 1.** Demographics of the sample

| Group of Ukrainian women (N = 43) | |
|---|---|
| Age (M ± SD; range) | 37.71 ± 10.71; 17–60 |
| Education (%) | |
| Secondary | 14.0 |
| Tertiary (bachelor's degree) | 2.3 |
| Tertiary (master's degree) | 76.7 |
| Tertiary (PhD.) | 7.0 |
| Marital status (%) | |
| Single | 16.3 |
| Married | 51.2 |
| Divorced 1 | 8.5 |
| Widowed | 4.7 |
| With a partner | 9.3 |

nationality. When asked about their national identity, 88% said Ukrainian, 7% said Ukrainian-Czech and 5% said Czech-Ukrainian.

For the purpose of the study, we designed a detailed interview supplemented by quantitative methods. Moreover, we measured the association of *adverse childhood experiences* (ACEs), the importance of the war to their identity and life (via the *Centrality of Events Scale* [CES]) with their experience of trauma (*PTSD Checklist – PCL-5*) using Spearman's correlation coefficient.

The interview consisted of 91 questions and lasted approximately 60–90 min. It was conducted in the respondents' native language by two Ukrainian psychologists during June–November 2022 in Brno, at the CEITEC. It focused on three broad variables: (1) prewar experiences, (2) war experiences and (3) adaptation to the Czech Republic. Some questions were open-ended, whereas other items required response scales.

Participants were asked about basic demographic data, information regarding close people, events preceding the war, the course of experiences with the war in Ukraine and psychological difficulties. In addition, they were asked to indicate their most stressful traumatic events, what effect the war had on them, their experience of living after immigration to the Czech Republic and their overall assessment of their own lives in both professional and personal spheres.

The interview included questions on ACEs (Kalmakis and Chandler, 2014), to which the respondent answers yes/no. Administering questions on ACEs is quick, getting specific answers is relatively easy and implementing them into interviews that inform mental health professionals is simple. We have included an eight-question version (Sacks and Murphey, 2018) and asked adult women about their experiences during the first 18 years of their lives. The total score ranges from 0 to 8. Higher scores represent a more intense rate of ACEs.

The interview also included the CES (Berntsen and Rubin, 2006). This method measures how central an event is to a person's identity and life story. Questions were asked in connection with the war in Ukraine. The total score ranges from 0 to 35. A higher score on this scale represents a higher degree of event centrality. In the original study, an internal consistency of. 88 was reported (Berntsen and Rubin, 2006).

Furthermore, we administered *PTSD Checklist – PCL-5* (Blevins et al., 2015). The PCL-5 is a 20-item self-report measure that assesses the 20 *DSM-5* symptoms of PTSD. The total score ranges from 0 to 80. A higher score is associated with a greater level of PTSD symptoms. A cut-off raw score is 38 for a provisional diagnosis of PTSD and this cut-off has high sensitivity (.78) and specificity (.98) (Cohen et al., 2015). In our study, the Ukrainian version was used (Agajev et al., 2016).

## Results

Respondents' experiences of the psychological trauma of warfare in Ukraine varied from hiding in a cellar for hours during air raids to spending weeks in occupied towns under systematic bombardment without water and electricity. At the time of the interviews, they had been in the Czech Republic for 3–42 weeks; the most common duration was 6 weeks (35% of the sample). On the day of the examination, on a five-point scale, 12% reported feeling excellent, 37% reported feeling very good, 33% reported feeling somewhere in between, 14% reported feeling rather bad and 5% reported feeling bad.

### Childhood adversities

The mean ACEs was 3.0 (SD = 1.5, range 0–6). Only one of the Ukrainian women had a score of ACE = 0 (i.e., no ACEs) out of the eight items. Thirty-three percent of the cohort had scores of 4 or more. When we asked about respondents' experiences in childhood (ACEs, first 18 years of life), we found that 74% of the cohort lived with a parent or caregiver who had divorced or separated from the family; 12% lived with a parent or caregiver who had died; no respondents lived with a parent or caregiver who was in custody or prison; 16% lived as a child in a household with someone who was mentally ill, suicidal or severely depressed for more than a few weeks; 28% lived as a child with a person who had alcohol or drug problems; 44% lived with a parent, caregiver or other household member who was violent to others (slapped, hit, kicked beat); 53% were victims of or observed violence in the neighborhood and 70% experienced material hardship rather often or very often.

### War and psychological problems

The mean PCL-5 score was 39.2 (SD = 14.7, range 5–69). The majority of participants (79%) had PCL-5 scores of 38 or higher, which is the cut-off score for a provisional diagnosis of PTSD (Cohen et al., 2015).

The mean of CES embedded in the survey was 28.7 (SD = 7.2, range 11–35). Moreover, CES showed that 49% of respondents believed that the event had become part of their identity, 53% said that the event had become an important aspect in how they understand themselves and the world, 49% said that the event had become a central part of their life story, 58% said that the event had influenced the way they think about and live other experiences and 60% said that the event had changed them permanently (score 5, strongly agree).

When self-assessing their health status, 28% of respondents reported that they had seen a psychiatrist as an outpatient at some point in their lives, 2% had been hospitalized in a psychiatric clinic and 7% of the cohort had been diagnosed with post-traumatic stress disorder (PTSD). In the past, 70% of the respondents took medicines "for nerves," currently 37% take these medicines and 33% take herbs "for nerves." Lifetime drug use was reported by 2% of the respondents. When asked about health complications, 14% mentioned heart-related problems, and a serious physical illness for which they were being treated was also mentioned by 14%. Lifetime alcohol problems were admitted by 14%. COVID infection was reported by 84% of the respondents.

Regarding wartime experiences, 51% of the sample reported that they had to hide during the war (e.g., from bombing). Enough food and drink during the war was reported by 72%. Shooting was experienced by 74%, 81% experienced bombing, 7% suffered injuries, 21% saw dead or wounded people, 12% had direct experience of combat, 5% had been employed as a professional soldier and 12% were captured or held hostage during the war. In addition, 7% said they experienced torture and 77% said the war was the most difficult event in their lives. The war is remembered every day by 60%, 28% often, 9% sometimes and 2% hardly at all. In 44% of the sample, war appears in their nightmares.

When asked about the most difficult event during the war, respondents mentioned, for example, staying underground in Mariupol (this city was hit hard by air raids and artillery strikes, communication was cut off and food and water supplies were also cut off), being shelled in the same place for a month, experiencing the death of a friend who was shot near Bucha, having a father who

was fighting, having the fear of dying, hiding from bombs in the subway in Kharkiv, leaving Kiev during bombing, seeing a man killed and not being able to bury him, knowing neighbors' houses were bombed, leaving with children (one child has autistic disorder and thus it was difficult to be in a confined space with other people), experiencing the occupation of the area, leaving Ukraine alone during shelling, experiencing Mariupol being razed to the ground, seeing their own house burn after the bombing, witnessing the start of the bombing of Mariupol, not understanding how to protect her child (one respondent); driving past a military convoy and thinking they were going to be killed, experiencing crowding and explosions at the train station, experiencing bombing and seeing their son-in-law being killed in front of their eyes. Other answers were also war-related.

Concerns about loved ones were featured in the responses, with some respondents stating, for example, that "I thought my parents were dead." The combination of challenges both in war and private life were noted as well, e.g., "war and living with one man," where respondents connect the complexities of a relationship with war and seeing a bloodied father who had been hit on the head by his mother's new husband. Forced family separation was a theme. For example, respondents gave answers such as, "I can't see my parents, they are in the occupied territory." Other difficult situations regarding family separation were also noted, for example, the eldest son did not want to leave Ukraine, relatives were in the occupied area in Bukhia and relatives and friends were staying in areas that were occupied. Another theme was adaptation to a new life situation. The answers given included situations such as the challenges of moving to another country, forced emigration abroad, fear for their child and his or her future, a missing husband or moving from one's beloved city. There were also exclusively private themes, for example, postnatal depression, relationship with one's husband, relationships that had escalated during the war or one's mother's death.

Regarding their own family during the war, 93% of respondents said that someone in their family or close to them had to hide in a shelter during the war, 21% said that someone in their family had been captured or held hostage, 30% said that someone in the family or close people had been injured during the war, 28% said that someone in the family or close people had direct experience of combat, 9% said that someone in the family had been killed during the war and 35% said that a neighbor had been killed during the war.

When asked about their own development due to the war, 79% of the sample said that the war had changed them, 65% said that it was easy to talk about the war with family and 74% said that it was easy to talk about the war with strangers. When asked about changes in their behavior caused by the war, 51% reported being more aggressive, 74% reported more emotional lability, 88% reported more anxiety and 49% reported increased withdrawal from life. When questioned, they also reported positive changes, for example, 47% reported increased friendliness and 53% reported increased tolerance of others.

### Stay in the Czech Republic

In the households in which they lived in the Czech Republic, three persons in the household were predominant (37%), 23% reported two persons, 16% reported four persons, 14% of the respondents lived alone, 7% reported six household members and 2% of the sample reported seven household members. Of the respondents, 34% were currently unemployed (but they still had a financial reserve), 72% reported below-average income relative to the average income in the Czech Republic, 11% reported average income and 2% reported no income. At the time of the interviews, 67% of respondents in the Czech Republic were receiving financial support.

Of the women surveyed, 72% have children. In some cases, the children were together with their mothers, in other cases, they were separated (e.g., the son stayed in Ukraine, the son stayed in a sports school in Hungary) or the mother and the children are in the Czech Republic but live separately (e.g., the daughter and the son live in Brno but separately, etc.).

When asked how the war experience had affected their children's upbringing, respondents' reports included the following themes: changes in the level of child protection, such as overprotection of offspring, restraining daughters from doing anything wrong, overprotection for safety, less shouting at their children and increased permissive parenting. Anxiety in parenting was also emphasized. The responses included constant tension and desire for their children's protection, constant fear for their son, feelings of anxiety when in contact with their children and changes in parenting. For example, parenting priorities have changed, such as the tendency to give less orders than before, having more respect for the family, giving more attention to their children or offering more protection for their children. The parenting theme extends to more general parental changes with direct responses included here, such as, "I have become more demanding and irritable," "I have become stricter," "he (the child) is nervous." Additionally, parent respondents reported difficulties with adaptation, which included responses such as their child has poorly adapted to school, relationships with other children have deteriorated or there are many arguments at home. Positive changes are also mentioned, which include responses such as, "relationships have become closer," "we talk a lot about the war – about feelings," "we support each other," "I have started to allow everything," "I have not been so strict on her behavioral disorders," "I have stopped demanding good studies," "the main thing is that they are alive," "I have become more responsible," "I am teaching my child to appreciate what she has." The majority of respondents (65%) expressed a belief that their children would have been different in character if the war had not happened, and 53% said that the war had affected their communication with their children.

When asked about their stay in the Czech Republic, 79% of the respondents said they perceived a language barrier, 60% had made friends in the Czech Republic and 23% were working in a job that matched their education. When asked about satisfaction in the Czech Republic, 51% said they were definitely satisfied in the Czech Republic, 37% said they were rather satisfied in the Czech Republic, 5% said they were rather dissatisfied in the Czech Republic, 5% said they were definitely dissatisfied in the Czech Republic and one woman did not answer.

With regard to the future direction of their lives, 47% of the respondents said they wanted to return to Ukraine, 26% said they wanted to settle long-term in the Czech Republic, 14% said they wanted to settle permanently in the Czech Republic, 2% said they wanted to settle long-term in a country other than the Czech Republic and 2% said they wanted to settle permanently in a country other than the Czech Republic (the rest of the sample – 9% – did not answer this question).

When asked about success at work, 58% reported a feeling of success at work, 7% reported "rather success," 5% gave the answer "somewhere in between," 2% reported "rather failure," and 28% reported a feeling of failure. When asked about success in their

**Table 2.** Relationship between PCL-5 total score and ACE and CES items (only significant results are shown)

| Method | Item | Item rating | Spearman's ρ, p |
|---|---|---|---|
| ACE | (During your childhood) did you live with anyone who was mentally ill or suicidal, or severely depressed for more than a couple of weeks? | Yes/no | .348, p = .022 |
| CES | I feel that this event has become part of my identity. | 1–5, 1 = completely disagree 5 = completely agree | .400, p = .008 |
| | I feel that this event has become a central part of my life story. | 1–5, 1 = completely disagree 5 = completely agree | .331, p = .030 |
| | This event has colored the way I think and feel about other experiences. | 1–5, 1 = completely disagree 5 = completely agree | .349, p = .022 |
| | This event permanently changed my life. | 1–5, 1 = completely disagree 5 = completely agree | .468, p = .002 |
| | This event permanently changed my life. | 1–5, 1 = completely disagree 5 = completely agree | .373, p = .014 |
| | This event was a turning point in my life. | 1–5, 1 = completely disagree 5 = completely agree | .333, p = .029 |

Abbreviations: ACE, adverse childhood experiences; CES, Centrality of Events Scale.

**Table 3.** Relationship between PCL-5 total score and interview items (Only significant results are shown.)

| Interview item | Item rating | Spearman's ρ, p |
|---|---|---|
| How do you feel today? | 1–5, 1 = excellent, 5 = very poor | .500, p < 0.001 |
| Would you say you coped well with the war experience? | 1–5, 1 = yes, 2 = rather yes, 3 = something in between, 4 = rather no, 5 = no | −.397, p = .008 |
| If you have children: would you say that the experiences of the war have influenced the upbringing of your children (e.g., overprotection of offspring, constant preparedness for emergencies…)? | 1–5, 1 = yes, 5 = no | .448, p = .013 |
| Do you think the war had an impact on the way you communicate with your children? | 1–5, 1 = yes, 5 = no | .501, p = .005 |
| Would you say that the war changed you toward more closed–mindedness? | Yes/no | .378, p = .014 |
| Overall satisfaction in the Czech Republic. | 1 = definitely satisfied, 2 = rather satisfied, 3 = rather dissatisfied, 4 = strongly dissatisfied | .326, p = .033 |

personal life, 53% reported success, 7% reported "rather success," 7% reported "somewhere in between," and 33% reported failure.

### Associations between PCL-5, ACE, CES and interview items

The correlation between PCL-5 and ACE was not significant (Spearman's ρ = .195; p = .209), similarly between CES and ACE (ρ = −.129; p = .409). A significant relationship, however, was found between PCL-5 and CES (ρ = .390; p = .010).

Additionally, we have analyzed the correlations between the trauma-related total score of the PCL-5 with individual items from ACE, CES and the interview. Significant results of ACE and CES items are shown in Table 2 (see below), and the results of the interview items are shown in Table 3 (see below). Only one item from ACE was related to PCL-5 total score (living with someone mentally ill, suicidal or depressed), while six items from CES were related to PCL-5 (ρ range from. 331 to. 468; *p*s < .05). Similarly, six items from the interview had correlations with PCL-5 total score (ρ range from. 326 to. 501; *p*s < .05).

### Discussion

The Ukrainian refugee wave was the result of the war with Russia that started in February 2022, and has affected countries that previously tended to reject refugees, such as Poland, Slovakia, Hungary and the Czech Republic, yet nonetheless accepted Ukrainian refugees during this time. The aim of this study was to explore and share the experiences of several Ukraininan women refugees in one of these countries, the Czech Republic.

For comparative context, studies of the Polish experience with refugees showed the following barriers to adaptation for Ukrainian women refugees: (1) language barrier, (2) temporality/liminality of the situation of exile, (3) trauma and psychological stress, (4) limited access to childcare, (5) stereotyping, (6) lack of networks and (7) high expectations of the host community toward refugees (Synowiec, 2022). Comparing these variables with our data, we have some overlapping ones and will specify our findings briefly here: (1) 79% of the sample reported perceiving a language barrier, (2) approximately half (51%) reported being definitely happy in the Czech Republic and approximately half (47%) wanted to return to Ukraine, (3) 79% reported that the war had changed them, (3a) of these, 88% reported higher levels of anxiety, (3b) 70% had never taken medication for psychological problems during their lives and (3c) 37% are now taking medication for psychological problems, (4) 53% of respondents reported that experiences of the war influenced the upbringing of their children, resulting from family separation, (5) we did not specifically address questions about stereotyping, but 10% reported that they were rather or definitely dissatisfied in the Czech Republic (one respondent reported that she had housing problems, lived in a hostel, could not sleep because of the noise, and was forced to be near people she considered enemies and who supported the Russian aggressor), (6) we did not directly address issues of lack of networks and (7) we did not directly address issues of attitudes of Czech residents toward refugees.

## Childhood

When asked about the respondents' childhood experiences (first 18 years of life), we found that 74% of the Ukrainian cohort had lived with a parent or caregiver who had divorced or separated from the family. For reference, in the Czech cohort, the percentage of women with this experience was 23.3% (WHO, 2023; Velemínský Sr et al., 2017, 2020). In the Ukrainian cohort, 12% lived with a parent or caregiver who had died and no respondents lived with a parent or caregiver who was in detention or prison, whereas in the Czech cohort, the latter category was 1.4%. In the Ukrainian cohort, 16% had lived as a child in a household with someone who was mentally ill, suicidal or severely depressed for more than a few weeks; in the Czech cohort, mental health illness was found in 13% of the corresponding cases and suicide in 4.2%. In the Ukrainian cohort, 28% of women lived as a child with a man who had alcohol or drug problems; in the Czech cohort of women, 15.4% reported alcohol in the household and 4.9% reported drugs. In the Ukrainian cohort, 44% lived with a parent, caregiver or other household member who behaved violently toward others (slapped, hit, kicked beat); in the Czech cohort, this situation was 21.3%.

In the Ukrainian sample, 53% of women had been victims of violence or observed violence in their neighborhood, and 70% had experienced material hardship rather often or very often (e.g., it was difficult for the family to cover the cost of food or accommodation). Also in the Ukrainian cohort, there was one respondent without ACEs (ACE = 0), but 33% of the cohort had scores of 4 or more. In the Czech study (Velemínský Sr et al., 2017), 38% of respondents had 0 ACEs and 9.9% of respondents reported experiencing four or more types of negative childhood experiences. In the Czech study, 17.1% reported physical abuse and 22% reported domestic violence as a witness. Material hardship was not observed in the Czech study. To summarize, there is a marked difference in the rate of childhood adverse experiences between the two countries, which is higher in the Ukrainian cohort.

One of the ACE items was related to the current level of trauma as measured by PCL-5 ("Did you live with anyone who was mentally ill or suicidal, or severely depressed for more than a couple of weeks?"), and was consistent with previous findings that ACEs and adulthood trauma (like war experiences) are risk factors for PTSD and CPTSD (Haim-Nachum et al., 2022; Redican et al., 2022), with the possible neurobiological explanation that ACEs negatively influence early post-trauma thalamic volumes which, in turn, are negatively associated with PTSS (post-traumatic stress syndrome) in survivors who develop PTSD (Xie et al., 2022), but also to other psychological disorders (van der Feltz-Cornelis et al., 2019; McCutchen et al., 2022). Children from families where the parents were separated or divorced suffer from PTSD at a higher rate (Graham-Bermann and Levendosky, 1998). This relationship was not demonstrated in our study, but the rate of separation or divorce in the original families was high, 74%.

The data show that the Ukrainian respondents in our study (43 women) reported ACEs to a greater extent than Czech women. The high rate of material hardship (70%) in the Ukrainian cohort is also very high compared to the clinical experiences of the corresponding Czech cohort. It is evident that the "starting" conditions of these Ukrainian women are burdened with circumstances that make it difficult to be ready to adapt to the challenging conditions, and thus maladaptive behavior is relatively high. For example, according to the estimates of the psychologists who surveyed the sample of Ukrainian women, approximately 30% drink alcohol every day (which corresponds approximately to the percentage – 28% – of Ukrainian women who reported that as children they lived in a household with a person who had alcohol or drug problems).

## Traumatization

Previous research indicates that Ukrainian women and young adults under 30 years of age are significantly more likely to suffer from mental health problems (Prokop et al., 2022). Ukrainian women have experienced situations that most individuals who do not experience war will not experience. Among other experiences, female respondents experienced shooting (74%) and bombing (81%), while 7% suffered injuries and 21% saw dead or wounded people. Even at the time of the interviews, one respondent was shaking. She was very nervous and reported not being able to find her way or remember her schedule at work. She also reported waking up in the middle of the night, having panic attacks and not feeling joy or satisfaction in her life.

The mean PCL-5 score for the participants was 39.2 (SD = 14.7). Seventy-nine percent of the cohort had a PCL-5 score of 38 or higher (the cut-off raw score is 38 for a provisional diagnosis of PTSD, Cohen et al., 2015). Therefore, it can be inferred that almost four-fifths of the cohort could also have a clinical diagnosis of PTSD. Increased levels of traumatic symptoms may be reflected in the care of their offspring, especially if other stressors are present (e.g., separation of families, need for acculturation, unfinished war, concerns about loved ones in Ukraine). In one study (Chudzicka-Czupała et al., 2023), 57.2% of the Ukrainian participants (general population living in Ukraine, data from an online survey collected March 8 to April 26, 2022) exceeded the IES-R cut-off for PTSD (score > 32). Similar estimates suggest that approximately one-third of refugees pass the diagnostic threshold for PTSD, depression and anxiety (Mesa-Vieira et al., 2022). Data from an online survey of Ukrainian respondents using the six-item International Trauma Questionnaire (data from and April 7 to 15, 2022; Ben-Ezra et al., 2023) show that 30.8% (616 respondents) met the criteria for an

elevated risk for PTSD. A study of Ukrainian refugees in Germany found severe psychological distress in 46% of women compared to 20% of men according to GHQ-12 results (Buchcik et al., 2023). Our data show similarly high levels of traumatization.

Six out of seven items from the CES were significantly related to traumatization as measured by the PCL-5, with items relating to the impact of the war on identity, the overall impact of the war on their life story, the lasting change it brought to their life, thinking about the impact of the war on the future, and seeing the war as a life turning point (all $p < .05$). The results suggest that the CES rate represents the level of traumatic symptoms as measured by the PCL-5. At this rate, trauma is a reference point for everyday inferences (Berntsen and Rubin, 2006), and intruding memories influence one's thinking about the future.

The topic of leaving Ukraine also came up in the interviews as a particularly significant stressor. One respondent stated that, "*We didn't have enough food on the way, I didn't expect the journey to take so long. At one point I realized that I had nothing to feed my child. I also did not eat for several days.*" Another respondent said that the journey out of the country "*was difficult; it took 6 days and she and I and my daughter did not have enough food on the road, it was very scary.*" Other researchers also reported respondents' hardships of crossing the border, long waiting times and the harsh conditions during their long journey (Oviedo et al., 2022). The difficulty of adaptation to the host country is certainly high, but on the other hand, internally displaced people who have been displaced by the ongoing conflict also have significantly higher levels of PTSD, for example, compared to urban-dwelling people who were not displaced (Johnson et al., 2022). Although in the Czech Republic and elsewhere we see the problem of refugees crossing national borders, in Ukraine, many citizens have had to leave their homes even though they remained in their own country. This corresponds to the dilemma of Ukrainians when deciding whether to stay in their home country or to leave, and whether to stay in the host country or to come back home.

### Past, present and future

In the interview items that correlated with PCL-5, items focusing on the present (e.g., how do you feel today, satisfaction with being in the Czech Republic) intersected with the past (e.g., asking about ACEs) and the future (e.g., asking about raising children, the impact of war on the future from CES). It can be inferred that trauma changes the overall attitude toward the course of life, encompassing its past, present and future. As some authors state, "In such cases, a highly negative, unpredictable and probably rare event will influence the attribution of meaning to other more mundane events as well as the generation of expectations for future events" (Berntsen and Rubin, 2006, p. 220). It is also possible that this perspective may change over time due to the dimensions of the trauma itself, one's own adaptation or the result of treatment, as one's life story is a narrative representation of the past, present and expected future (McAdams, 2001).

While a common defense mechanism of survivors of World War II was adhering to the *conspiracy of silence* (e.g., Danieli et al., 2016; Cohn and Morrison, 2017), 65% of Ukrainian women reported that it was easy to talk about the war with family and 74% reported that it was easy to talk about the war with strangers. Some women even described positive changes; for example, 47% reported increased friendliness and 53% increased tolerance toward others. One respondent reported a need to adapt quickly to the new conditions in the Czech Republic: "*When I moved here, I grew up in an instant. All the responsibility is on me now. I work hard and that's how I save myself. I've never worked as hard as I'm working now.*"

Why is not the defense mechanism of the conspiracy of silence also repeated for Ukrainian women? There may be several reasons for this: (1) the war in Ukraine does not yet meet the standards of genocide; (2) it is also relatively easy to escape from Ukraine compared to Hitler's Germany; (3) Ukraine has a high level of support in the Western world and the West is active in the armed conflict, including supplying weapons, logistical support and full acceptance of Ukraine as an invaded country and (4) there are now more accessible ways to connect and share trauma.

### Stay in the Czech Republic

The number of emigrants who wish to remain in the host country naturally varies according to the state of war and their acculturation. As we know from another study, maintaining communication with separated loved ones is a principal element for coping and resilience (Oviedo et al., 2022). Ukrainian women and their children have spouses and relatives who remained in their home country. The experience of separation combined with anxiety negatively affects the mental state of refugees (Synowiec, 2022). Family reunification or separation are major transformations that determine whether an individual stays or returns to the host country, with the quality and stability of ties in the home country as the foundation of the decision. Observations and interviews with a group similar to ours – 30 refugee women with young children on the Polish-Ukrainian border (Synowiec, 2022) – showed that the women spoke only Ukrainian and/or Russian. All reported a deterioration in their own (and family's) socio-economic status (our data show that only 23% of Ukrainian women refugees in the Czech Republic have a job corresponding to their education). In one study in the Czech Republic, approximately 14% of the respondents were planning to return home (Prokop et al., 2022), and at the end of 2022, 39% of children were well integrated into a group of Czech children, according to their parents (Prokop et al., 2022). With regard to the future direction of their lives, 47% of the respondents said they wanted to return to Ukraine, while 40% said they wanted to settle in the Czech Republic permanently or in the long term. Similar figures are also available in the neighboring country – about 30% of war refugees from Ukraine want to stay in Poland permanently (data from March to June 2022, Synowiec, 2022).

In June 2022, MoLSA Minister Marian Jurečka stated that over 70,000 refugees from Ukraine with humanitarian visas and labor permits have found jobs and started working in the Czech Republic. In January 2023, the Czech authorities announced that nearly 100,000 refugees from Ukraine had found work in the Czech Republic and paid roughly CZK 8 billion in taxes and more than CZK 4 billion in health insurance to the state in 2022 (Czech Press Office, 2023).

### Limitations

The representativeness of our data is limited by the small sample size and the focus on women. We did not account for other factors that may influence post-traumatic stress levels, for example, patriotic attitudes, which have been studied during the current wartime (Hamama-Raz et al., 2022). PTSD accompanied by post-traumatic stress symptoms are just one of the possible negative effects of war. They include – among others – panic disorders, rumination, conduct disorders and lower levels of well-being (Elvevåg & DeLisi, 2022).

## Conclusions

Ukrainian women were interviewed shortly after the start of the war and their arrival in the Czech Republic, having spent only 3–42 weeks in the country, most often a duration of 6 weeks. Although more participants felt better than worse on the day of the examination, it is evident that the psychological demands of coping with the situation of emigration are high. Within a short period of time, they experienced familial separation and concrete loss of life. For example, approximately one-third (35%) of the participants lost neighbors killed in the war. Most women experienced bombing and shooting during the war. High levels of distress are also indicated by high levels of trauma symptoms in our study, which on average exceeded the cut-off score for probable PTSD. Yet 60% of women had made friends in the Czech Republic, showing the positive potential for resilience even under dire circumstances.

The demands of current functioning under conditions of emigration may be highlighted by the fact that participants had experienced ACEs (33% of the cohort had ACEs score of 4 or more). Compared to the relative material comfort of the Czech Republic, 7 out of 10 participants experienced material hardship rather often or very often during their childhood, which is unusual in the conditions of the host country. On the other hand, our results show that there is a closer relationship between traumatization (PCL-5 outcome) and how central an event is to a person's identity and life story (CES outcome) than between traumatization and early childhood experiences (ACEs). Almost four-fifths of the women said that the war had changed them. More than half of the women believed that the war had changed them permanently.

Previous generations did not have such a relatively easy opportunity for therapeutic dialog with the helping professions, such as social workers, psychologists and psychiatrists as people do today. Despite these conveniences that emerged during the second half of the 20th century, the refugee situation remains extremely challenging in the 21st century. It is hoped that the interviews we conducted with the respondents can have an integrative effect for them in addition to the research purposes and can sometimes lead to catharsis, exploration, integration of emotions and thoughts and better regulation of emotions, giving meaning through different forms of reappraisal and search for meaning (Dubovská et al., 2022).

**Open peer review.** To view the open peer review materials for this article, please visit http://doi.org/10.1017/gmh.2024.7.

**Data availability statement.** Data are available on request.

**Acknowledgment.** This project is supported by grant NU22-04-00661.

**Author contribution.** Conceived and designed the analysis – I.R., M.P., S.B., T.Y. and M.F. Collected the data – M.F., S.B. and T.Y. Contributed data or analysis tools – M.P., R.H. and P.W. Performed the analysis – R.H. Wrote and/or edited the paper – I.R., M.P., S.B., T.Y., M.F., R.H., E.S. and P.W.

**Financial support.** This research was supported by Central European Institute of Technology (CEITEC), Centre for Neuroscience at Masaryk University in Brno.

**Competing interest.** The authors declare no competing interests.

**Ethics statement.** This research follows the ethical recommendations for the conduct, reporting, editing and publishing of scientific research. It was approved by the ethical review board of the Central European Institute of Technology (CEITEC) EKV-2021-076.

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
