## [Reviewer Report]

I have reviewed this paper. My concern is methodology. I find the correlation table is atypical. I am not sure how questionnaire or interview cannot correlate with an item. It is good to prepare a table on demographics. The authors should consult a statistician to check analysis method.

The authors should refer to previous studies on this topic such as:

Chudzicka-Czupała A, Hapon N, Chiang SK, Żywiołek-Szeja M, Karamushka L, Lee CT, Grabowski D, Paliga M, Rosenblat JD, Ho R, McIntyre RS, Chen YL. Depression, anxiety and post-traumatic stress during the 2022 Russo-Ukrainian war, a comparison between populations in Poland, Ukraine, and Taiwan. Sci Rep. 2023 Mar 3;13(1):3602. doi: 10.1038/s41598-023-28729-3. PMID: 36869035; PMCID: PMC9982762.

---

## [Reviewer Report]

There are lots of grammatical errors, run-on sentences, and mis-use of semicolons. The paper lacks topic sentences as well which confuses the reader. Abstract is written well, with good introduction and concluding statements. Apart from grammatical errors, the methods section was written well, the introduction was written well, the analysis appears reasonable. Discussion appears reasonable, although sentences are extremely long and run-on. As for the table, it is very confusing and should likely be separated into two or three different categories of information. Interpretation of the table is very difficult. I recommend revising the paper for grammatical corrections and writing style changes.